# Neutrophil Extracellular Trap Formation Correlates with Favorable Overall Survival in High Grade Ovarian Cancer

**DOI:** 10.3390/cancers12020505

**Published:** 2020-02-21

**Authors:** Besnik Muqaku, Dietmar Pils, Johanna C. Mader, Stefanie Aust, Andreas Mangold, Liridon Muqaku, Astrid Slany, Giorgia Del Favero, Christopher Gerner

**Affiliations:** 1Department of Analytical Chemistry, Faculty of Chemistry, University of Vienna, 1090 Vienna, Austria; besnik.muqaku@univie.ac.at (B.M.); johanna.c.mader@univie.ac.at (J.C.M.); liridon.muqaku@gmail.com (L.M.); astrid.slany@univie.ac.at (A.S.); 2Department of Surgery and Comprehensive Cancer Center, Medical University of Vienna, 1090 Vienna, Austria; dietmar.pils@univie.ac.at; 3Department of Obstetrics and Gynecology, Comprehensive Cancer Center (CCC), Medical University of Vienna, 1090 Vienna, Austria; stefanie.aust@meduniwien.ac.at; 4Department of Internal Medicine II, Division of Cardiology, Medical University of Vienna, 1090 Vienna, Austria; andreas.mangold@meduniwien.ac.at; 5Department of Food Chemistry and Toxicology, Faculty of Chemistry, University of Vienna, 1090 Vienna, Austria; giorgia.del.favero@univie.ac.at

**Keywords:** ovarian cancer, neutrophil extracellular traps, mass spectrometry, proteomics, metabolomics, eicosanoids

## Abstract

It is still a question of debate whether neutrophils, often found in the tumor microenvironment, mediate tumor-promoting or rather tumor-inhibiting activities. The present study focuses on the involvement of neutrophils in high grade serous ovarian cancer (HGSOC). Macroscopic features classify two types of peritoneal tumor spread in HGSOC. Widespread and millet sized lesions characterize the miliary type, while non-miliary metastases are larger and associated with better prognosis. Multi-omics and FACS data were generated from ascites samples. Integrated data analysis demonstrates a significant increase of neutrophil extracellular trap (NET)-associated molecules in non-miliary ascites samples. A co-association network analysis performed with the ascites data further revealed a striking correlation between NETosis-associated metabolites and several eicosanoids. The congruence of data generated from primary neutrophils with ascites analyses indicates the predominance of NADPH oxidase 2 (NOX)-independent NETosis. NETosis is associated with protein S100A8/A9 release. An increase of the S100A8/CRP abundance ratio was found to correlate with favorable survival of HGSOC patients. The analysis of additional five independent proteome studies with regard to S100A8/CRP ratios confirmed this observation. In conclusion, NET formation seems to relate with better cancer patient outcome.

## 1. Introduction 

The tumor microenvironment, including the immune system, may have an important impact on tumor progression and treatment response [1,2]. Focusing on tumor–stroma interactions, we are applying molecular profiling via mass spectrometry (MS), especially by the combination of proteomics, lipidomics, and metabolomics [3,4,5,6]. This approach supports the analysis of pathomechanisms and related adaptative responses such as inflammation in great detail [7,8]. With regard to high grade serous ovarian cancer (HGSOC), it allowed us to identify predictive marker profiles [9] as well as novel drug targets [10]. A distinct role of the immune system affecting the kind of tumor spread of ovarian cancer was described by us already previously [11].

High grade serous ovarian cancer still represents a major clinical challenge, the overall survival has only slightly improved within the last decades [12]. Peritoneal tumor spread and a unique tumor microenvironment provided by accumulated peritoneal fluid (ascites) characterize this disease. A classification based on macroscopic features distinguishes two types of peritoneal tumor spread in HGSOC [13]. The miliary type metastases appear as widespread and millet sized lesions, whereas non-miliary metastases are bigger in size, fewer in number and consist of exophytically growing implants. The first one is characterized by worst prognosis, increased systemic inflammation and little adaptive immune reactions in contrast to non-miliary which is probably spread via circulation and is more infiltrated by cells of the adaptive immune system [11,12,13,14].

Among the members of the innate immune system, neutrophils belong to the first responders to infections, injury or damage-associated molecular patterns (DAMPs) [15,16] and may eventually get activated by hypoxic conditions of the tumor microenvironment, as characteristic for the peritoneum in HGSOC [17,18]. Neutrophils exert their functions through phagocytosis, degranulation, or release of neutrophil extracellular traps (NETs) [19], a physiologic way of cell death called NETosis [17,18,20]. NETs are DNA structures associated with proteins such as histones and others with antibacterial activity, including neutrophil elastase (ELANE), myeloperoxidase (MPO), cathepsin G (CTSG), and lactoferrin (LTF) [20,21]. Under normal conditions, after successful clearance of an infection, a resolution of inflammation is initiated. However, malignant pathological conditions may be accompanied by chronic and apparently disorganized inflammatory response [17,18,22]. While immune therapy aiming at fostering immune responses against cancer cells has revolutionized cancer therapy [23], we have also observed that immune cells may have strong tumor-promoting capabilities [24]. Actually, apparently conflicting data are available regarding harmful [25,26,27] or desirable [28,29] effects of neutrophils in ovarian cancer.

The present study focuses on the potential contribution of neutrophils to mechanisms of tumor spread and overall survival in HGSOC patients. Integrating MS-based multi-omics analysis to ascites samples collected from HGSOC patients, we demonstrate the involvement of neutrophil-derived effector molecules in the pathogenesis of this disease and present a marker profile indicative for the neutrophil status in relation to systemic inflammation, which seems to be predictive for overall survival also in case of other tumor entities.

## 2. Materials and Methods

Ascites fluid was collected from HGSOC patients at the Medical University of Vienna with ethical approval from The Ethics Committee of the Medical University of Vienna and Vienna General Hospital (AKH), nos. 366/2003 and 793/2011. Metabolomics and Luminex-based cytokinomics data of cell-free ascites, immune cells composition data of ascites and transcriptomics data of tumor tissue were published before [11]. To the 25 cell-free ascites samples (11 miliary, 7 non-miliary, and 6 unknown) from the same sample pool, we applied MS-based proteomics and eicosadomics analysis. Additionally, we measured cell-free ascites samples collected from five patients with liver cirrhosis. A co-association network analysis was implemented using all multi-omics data. Further, in vitro experiments with neutrophils isolated from healthy donors were performed. A dextran protocol was implemented for the isolation of neutrophils from blood samples. After isolation, neutrophils were immediately treated with phorbol 12-myristate 13-acetate (PMA, Sigma-Aldrich, Vienna, Austria) and ionomycin (Sigma-Aldrich, Vienna, Austria). Mitochondria and endoplasmic reticulum were isolated from neutrophils by using a sucrose gradient. The experiments with neutrophils were approved by the Ethics Committee of the Medical University of Vienna (number: 1947/2014). Four different software packages were implemented to evaluate the data generated from three mass spectrometric instruments [30,31] (Appendix A).

An extended description of the material and methods can be found in the Appendix A. Briefly, for the proteomics analysis, ascites samples were depleted, while the supernatant samples of neutrophils were treated overnight with ethanol for protein precipitation. The protein samples were then digested with trypsin and analyzed by liquid chromatography (LC)-coupled MS analysis. The untargeted shotgun proteomics analysis with ascites samples was conducted on a QExactive Orbitrap instrument (Thermo Fischer Scientific, Waltham, MA, USA USA) coupled with an UltiMate 3000 RSLC nano system (Dionex, Sunnyvale, CA, USA). The targeted analysis was developed for the measurement of selected proteins in cell supernatant samples and was implemented on a QQQ6490 triple quadrupole instrument (Agilent, Santa Clara, CA, USA) coupled with a Chip-nano-LC system(Agilent) for online peptide separation. After removing the proteins by ethanol precipitation, the protein-free sample was used for isolation of eicosanoids by following a solid-phase extraction protocol. Eicosanoids were separated by using an HPLC chromatographic system (Agilent). The untargeted eicosadomics analysis of ascites samples was performed on a QExactive instrument (Thermo Fischer Scientific, Waltham, MA, USA), and a QQQ6490 triple quadrupole instrument (Agilent) was used for targeted analyses of eicosanoids in cell supernatant samples. The metabolomics analysis was performed using the AbsoluteIDQ p180 kit from Biocrates (Biocrates, Innsbruck Austria).

## 3. Results

### 3.1. Evidence for Activated Neutrophils in Ascites of Non-Miliary Type of Tumor Spread

Ascites samples were obtained from 18 patients as described previously [9] and classified according to macroscopic features upon surgery into miliary (11 patients) and non-miliary (7 patients). Figure 1A indicates proteins caracteristic for NETs according to Brinkmann et al. [20] which were found up-regulated in non-miliary ascites samples (Appendix A). This observation indicates functional activation of neutrophils in patients with non-miliary tumor spread. Anti-oxidant proteins such as glutathione S-transferase P (CSTP1), glutathione S-transferase omega-1 (GSTO1), and glutathione synthetase (GSS) were also found up-regulated in non-miliary samples (Figure 1A). In addition, the calprotectin constituents S100A8 and S100A9 were found up-regulated in these samples. As the inflammation marker calprotectin is derived from activated phagocytes, it is linked with local inflammation [32,33]. In contrast, the liver-derived systemic inflammation markers C-reactive protein (CRP) and serum amyloid A-1 protein (SAA1) were found up-regulated in the miliary samples (Figure 1A).

### 3.2. Evidence for Eicosanoids Class Switching in Neutrophils from Non-Miliary Samples

A comparative analysis of polyunsaturated fatty acids as well as their oxidation products performed by high-resolution mass spectrometry identified six eicosanoids 17-HDoHE, 12S-HETE, 15S-HETE, PGE2, and PGB2) significantly up-regulated in the non-miliary compared to the miliary ascites samples: (+/-)-14-hydroxy-4Z,7Z,10Z,12E,16Z,19Z-docosahexaenoic acid (14-HDoHE), (+/-)-17-hydroxy-4Z,7Z,10Z,13Z,15E,19Z-docosahexaenoic acid (17-HDoHE), 12S-hydroxy- 5Z,8Z,10E,14Z-eicosatetraenoic acid (12S-HETE), 15S-hydroxy-5Z,8Z,11Z,13E-eicosatetraenoic acid (15S-HETE), prostaglandin E2 (PGE2) and prostaglandin B2 (PGB2) (Figure 1B and Appendix A). Remarkably, these eicosanoids are products from the enzymes 15-lipoxygenase (15-LOX), 12-lipoxygenase (12-LOX), and cyclooxygenase (COX), whereas products of the 5-lipoxygenase (5-LOX) enzyme [34] were positively identified, but not significantly regulated (Figure 1B,C and Appendix A and Appendix A). As inflammatory stimulated neutrophils initially release mainly 5-LOX products, this finding suggested that neutrophils in the non-miliary samples had already switched to 12-LOX, 15-LOX, and COX-products known to be involved in the resolution of inflammation [22] (Figure 1C). The COX-product PGE2 (Figure 1B) is actually known to further promote eicosanoid class switching in neutrophils initiating a feed-back loop [22,35].

### 3.3. Co-Association Network Analysis Revealed a Strong Correlation of Six Metabolites with Eicosanoids 

In order to learn more about the pathophysiological processes occurring in ovarian cancer patients, a co-association network analysis was generated. For that, the proteomics and eicosadomics data presented above were complemented with previously published data (metabolomics, transcriptomics, cyto/chemokine analyses, and FACS data) [11]. The results show a molecular network signature with six metabolites (glutamate, aspartate, spermidine, spermine, taurine, and histamine) establishing a hub in the center of the network predominantly correlating with eicosanoids (Figure 2). All six metabolites showed significantly increased concentrations in non-miliary compered to miliary ascites samples [11] (Figure 2). Thus, a co-regulation of eicosanoids with metabolites was strongly suggested, motivating us to investigate the underlying pathomechanism. Remarkably, the ani-inflammatory cytokine interleukin-10 (IL-10) was found positively correlated with miliary samples.

### 3.4. Activated Neutrophils May Account for Most Molecular Alterations Observed in Ascites Samples 

To verify whether activated neutrophils might represent a plausible source of deregulated molecules in the non-miliary ascites samples, we performed in vitro stimulatory experiments with neutrophils isolated from healthy donors. Formation of NETs in a process termed NETosis is eventually resulting from strong neutrophil activation [18]. Following previous studies, isolated neutrophils were treated either with phorbol 12-myristate 13-acetate (PMA), inducing NOX-dependent NETosis, or ionomycin, inducing NOX-independent NETosis [36,37,38,39]. Proteome profiling demonstrated up-regulation of some NETs proteins (indicated in Figure 1) already after one hour of treatment, and up-regulation of most indicated NETs proteins three hours after treatment with PMA or ionomycin (Figure 3A). The effect obtained with ionomycin was apparently stronger compared to PMA. The metabolomics analysis of neutrophil supernatants, focusing on the six metabolites building a hub in the middle of the network, revealed up-regulation of spermine, spermidine, histamine, and taurine three hours after treatment (Figure 3B). This suggests that neutrophil activation might also be related to the variations in the metabolic profile [11] (Figure 2). Indeed, the positively charged polyamines have already been described to be released together with the NETs from activated neutrophils [40]. Furthermore, spermine has been described to attenuate mitochondrial swelling induced from high cytoplasmic calcium concentrations as caused by ionomycin treatment [41]. Thus, in order to verify if we could reproduce a similar response in vitro, we performed immunofluorescence experiments aimed to compare the subcellular localization of spermine/spermidine in neutrophils before and after activation. Apparently, the activation with PMA induced the enlargement of ER, stronger than with ionomycin, while the polyamines were found to co-localize with both ER and mitochondria upon treatment (Figure 3D). As anti-spermine antibodies cannot distinguish between spermine and spermidine, we extended this analysis with LC-MS analyses of sub-cellular fractions enriched in mitochondria and endoplasmic reticulum from untreated and stimulated neutrophils, respectively. Spermidine, but not spermine, was apparently strongly retained in the ER, upon induction with PMA. Both polyamines were found up-regulated in the mitochondrial fractions upon ionomycin treatment (Figure 3E). The biological functions of taurine, spermidine and spermine are related to antioxidant properties [42,43,44]. Upon inflammation- mediated oxidative stress, taurine may neutralize toxic oxidants generated from MPO by activated neutrophils [42]. Actually, neutrophils are known to contain very high concentrations of intracellular taurine [42]. Thus, the presently observed increased taurine concentrations may also be related to neutrophil activation resulting in NETosis in non-miliary ascites samples. Also, the results from eicosanoid analyses of neutrophil supernatants are consistent with the suggested role of neutrophils in ascites samples. All six significantly regulated eicosanoids (Appendix A) were found induced upon stimulation, with increasing values after one and three hours, respectively. While the 5-LOX products were found strongly induced upon one hour treatment, their abundance values were found consistently decreased after additional two hours (Appendix A). In contrast, the similarly induced 12/15-LOX and COX products did not decrease upon prolonged incubation. These results were more contrasting when using 0.01% FCS (Figure 3C) instead of 10% FCS, because the high background levels of eicosanoids detected in 10% FCS were affecting the analysis. The results of the eicosadomics analysis were reproduced independently using neutrophils isolated from additional five healthy donors (Appendix A).

In summary, the in vitro experiments support the interpretation that activated neutrophils may represent the primary source of the observed alterations of eicosanoids and metabolites in ascites samples. Generally, the ionomycin-induced effects on neutrophils showed higher similarities to the molecular patterns observed in the ascites samples when compared to the observed effects using PMA, pointing to a rather NOX-independent activation pathway *in vivo*.

### 3.5. Higher S100A8/CRP Ratios in Ascites Samples Positively Correlate with Overall Survival

In clinical practice, elevated levels of CRP are often associated with unfavorable disease progression [45]. S100 proteins are regulated differently and thus show rather poor correlations with CRP [46,47,48]. As CRP rather relates to systemic inflammation in contrast to S100 proteins, a ratio of these proteins may indicate to what extent inflammatory deregulation was becoming systemic and thus relates to overall survival. Higher values of S100A8/CRP ratio were found in non-miliary samples described to have better prognosis (Figure 4B). In order to compare the present data with a non-neoplastic but severe inflammatory disease, liver cirrhosis was included in the present considerations. Remarkably, the S100A8/CRP ratio was found at similar rates in patients with miliay tumor spread when compared to patients with liver cirrhosis.

### 3.6. NETs Proteins and S100A8/CRP Ratio May Serve as Prognostic Biomarkers

To collect additional evidence for neutrophil activation in another kind of tumor and its potential association with overall survival, we re-evaluated published proteomics data from our laboratory generated from the analysis of cerebral melanoma metastases [49]. Melanoma patients (n = 18) undergoing a MAPKi therapy were classified depending on progression-free survival (PFS) in good (PFS ≥ 6 months) and poor responders (PFS ≤ 3 months). Comparing these two groups, the present analysis revealed significant up-regulation of several neutrophil-specific proteins, labeled in orange in Figure 5A, in metastases isolated from good responders compared to those of poor responders. Actually, similar to one-hour neutrophils treatment (Figure 3A), here we observed no release of histones but higher levels of cathepsin G (CTSG), suggesting neutrophil activation not necessarily resulting in NETosis. Anyhow, the molecular signature of neutrophil activation was also associated with better prognosis for the melanoma patients. In addition, we re-evaluated proteomics data from other laboratories regarding chemotherapy-naive HGSOC patients [50]. Again, NETs proteins were found up-regulated in patients more sensitive to chemotherapy (n = 14, median disease-free survival or PFS = 1160 days) in comparison to chemo-resistant patients (n = 11, median PFS = 190 days) (Figure 5C). Intriguingly, CRP was not found significantly regulated in this study, similar as in the melanoma study. However, the S100A8/CRP abundance ratio was found decreased in the group of patients with poor outcome (in poor responders and patients resistant to chemotherapy, Figure 5B,D). This finding suggested that the S100A8/CRP abundance ratio could be regarded as prognostic biomarker for ovarian cancer as well as for melanoma. However, CT45 (cancer/testis antigen 45) protein was reported to represent the best possible prognostic marker for long-term survival in the ovarian cancer study [50]. Re-grouping the HGSOC patients in this study according to overall survival with OSD < 1000 (n = 10) and OSD > 1000 (n = 15) revealed that S100A8/CRP abundance ratio significantly stratified these groups (Figure 5F). While CT45 protein seems to better predict chemotherapy response, S100A8/CRP seems to better predict overall survival (Figure 5D,F). In line, the S100A8/CRP abundance ratio was again found significantly up-regulated in the group of patients with OSD > 1000 as calculated from proteomics data of metastatic tumors isolated from HGSOC patients combining the study by Coscia et al. [50] and a more recent study by Eckert et al. (Figure 5G,H, n = 36) [51] Kaplan–Meier analysis confirmed that patients with an S100A8/CRP abundance ratio equal or above a cutoff value of 3.038 showed a significantly longer overall survival time compared to patients with values below this cutoff (Figure 5I). NETs proteins and S100A8/CRP ratio were up-regulated in ovarian cancer patients with favorable outcome of another recent study [52] (Appendix A). Thus, we calculated the ratio of the two proteins in a total of six studies (Figure 1A, Figure 5A–G and Appendix A ) representing all kinds of variations based on different patient cohorts and methodological details. Indeed, the S100A8/CRP ratio was found significantly correlated with overall survival (n = 116, Figure 5J, Appendix A).

## 4. Discussion

An active and relevant role of neutrophils for tumorigenesis has been recognized with regard to several tumors [53]. However, whether neutrophils rather promote cancer development or contribute to immune reactions inhibiting cancer growth, or whether neutrophils subtypes [54] may account for conflicting data, is still a matter of debate. Actually, neutrophil-lymphocyte ratios are increasingly used as prognostic and predictive factors [55,56]. Here, with regard to HGSOC we have collected ample evidence that NETosis is associated with non-miliary tumor spread. NETosis as well as the activity of tumor-associated macrophages have been rather linked to the promotion of metastasis [57,58]. While non-miliary tumor spread is indeed characterized by more invasive growth, it is associated with better overall survival. Apparently, we are dealing here with contradictory observations.

The present data suggest a model which might account for several apparent discrepancies (Figure 6). The model comprises three levels: (1) Initiation of NETosis by hypoxic cell stress; (2) establishment of distinct mascroscopic features related to a specific biomarker profile due to NETosis and (3) modulation of the adaptive immune system by NETosis promoting improved overall survival.

### 4.1. Initiation of NETosis by Hypoxic Cell Stress

A substantial challenge for ovarian cancer cells in the peritoneum is caused by hypoxia [59] calling for metabolic adaptation [60] Hypoxia has been described to induce the down-regulation of BRCA1 expression resulting in an impairment of cell cycle checkpoint control and DNA repair mechanisms [61]. In clinical practice, this knowledge has motivated combinatorial therapies including PARP inhibitors, anti-angiogenics, immune checkpoint inhibitors, phosphoinositide 3-kinase (PI3K), protein kinase B (AKT), mammalian target of rapamycin (mTOR), WEE1, mitogen-activated protein kinase (MEK), and cyclin dependent kinase (CDK) 4/6 inhibitors [62]. Actually, increased incidence of cell death may be associated with stress resulting in the release of DAMPs and the establishment of an adaptive immune response [63]. The release of glutamate, presently found up-regulated in non-miliary ascites samples (Figure 2) has been associated with hypoxic stress of cancer cells [64] potentially causing the initiation of NETosis (Appendix A) [65]. In addition, the strong chemokine and NETs inducer interleukin-8 (IL-8) was found up-regulated with glutamate (Appendix A) and was observed to be up-regulated in non-miliary ascites [11]. Furthermore, IL-8 signaling pathway and IL-8 activated PI3K signaling pathway were identified as one of the most deregulated pathways [66,67]. As a result, NETosis may account for all major molecular alterations associated with non-miliary ascites comprising proteins, eicosanoids and metabolites (Figure 1 and Figure 6).

### 4.2. Establishment of Distinct Mascroscopic Features Related to A Specific Biomarker Profile Due to NETosis 

NETs releasing neutrophils eventually release PGE2, IL-8, matrix metalloproteinase-9 (MMP9), ELANE (Figure 1A) [68] and other tumor promoters plausibly accounting for a more invasive growth of neighboring tumor cells. An improved access to blood supply resulting from invasion supported by neutrophil-induced neoangiogenesis may account for larger metastases typically observed in case of non-miliary metastases [69,70]. In contrast to this scenario, ovarian cancer cells better coping with hypoxic stress would rather remain restricted to the peritoneum as observed in case of miliary forms of metastasis, associated with less growth capacity but improved resistance to anti-cancer drugs, less involvement of immune cells and thus decreased overall survival of patients. Additionally, the increased levels of IL-10 (Figure 2) in miliary samples may account for an inhibition of NETosis [71], thus supporting miliary kind of tumor spread. Indeed, neutrophils are known as main sources of proteins S100 A8 and A9 [33], which may thus serve as biomarkers for NETosis (Figure 3A).

### 4.3. Modulation of the Adaptive Immune System by NETosis Promoting Improved Survival

The involvement of neutrophils in carcinogenesis is well recognized [53] but their role in cancer therapies is still subject of controversial debate [72]. Elevated levels of peripheral neutrophils correlated with poorer clinical outcome in oropharyngeal cancer [73] and have been described to promote metastasis in a mouse lung cancer model [74] as well as human breast cancer [75]. Here, we described that NETs formation is associated with non-miliary metastasis and better overall survival. This may be due to a modulation of the adaptive immune system via recruitment of CD8^+^ T cells and suppression of regulatory T cells, which were found increased in miliary samples (Appendix A) [11] This observation may also be linked to higher levels of CD3+/CD8+ tumor-infiltrating lymphocytes found associated with mutated BRCA1/2 in HGSOC [76]. We suggest here that the eicosanoids which were found deregulated (Figure 1B and Figure 2) may represent strong effector molecules regulating the local immune status. Most importantly, the present data regarding the prognostic power of the S100A8/CRP ratio seem to suggest a beneficial role of activated neutrophils for cancer patients. This may actually look like data inconsistency or contradicting observations.

A closer look on the present data may resolve this apparent conundrum and put the role of neutrophils for cancer progression in a new context. It seems to us that it is not necessarily the presence or absence of neutrophils which makes the difference. Here we suggest that it is the functional state of neutrophils in situ which matters. The present data are highly consistent with NOX-independent activation of neutrophils in the peritoneum accompanying non-miliary metastases. The ascites samples actually do not show an up-regulation of LOX-5-derived products which have been made responsible for tumor-promoting effects of neutrophils [74], but do show increased levels of LOX and COX-products associated with the resolution of inflammation. Microparticles containing 14-HDoHE and 17-HDoHE, shed from neutrophils, may inhibit the pro-inflammatory action of activated macrophages [77] which have been demonstrated to promote tumor growth and metastasis [58]. The present in vitro studies with neutrophils isolated from healthy donors demonstrate that stimulation via NOX or via calcium release causes substantially different effector functions in neutrophils. While PGE2 released from neutrophils in the periphery may support invasiveness and metastasis [78] local PGE2 release within the peritoneum may stimulate the immune system to better elicit an appropriate immune response. Indeed, increased levels of CD8^+^ T cells were described to be associated with non-miliary metastasis (Figure 2) [11]. 

In conclusion, both cell activities as well as the localization of specific events seem to matter. Obviously, it is not trivial to seek blood-borne tumor markers reporting specific events confined to specific locations. However, the presently described ratio of S100A8/CRP might provide at least some insight in that regard. Any systemic activation of the immune system including the activation of neutrophils will result in the increase of the highly sensitive and disease-relevant biomarker CRP as demonstrated in countless studies. Increased CRP is typically associated with poor prognosis, in line with all reports regarding the systemic involvement of neutrophils. S100A8, as well as S100A9, are released by neutrophils upon local NETosis (Figure 3A). Thus, it is the balance between local inflammation and systemic inflammation which may account for the apparent predictive power of S100A8/CRP ratio for improved survival.

## 5. Conclusions

The present data demonstrate a strong influence of NETosis on the local microenvironment accompanying non-miliary metastasis in HGSOC. The consequences involving the immune system may account for the improved overall survival reported for non-miliary metastases. A new marker profile, the S100A8/CRP ratio was found to correlate with improved survival and may report the individual balance between local and systemic activation of the immune system.

## Figures and Tables

**Figure 1 cancers-12-00505-f001:**
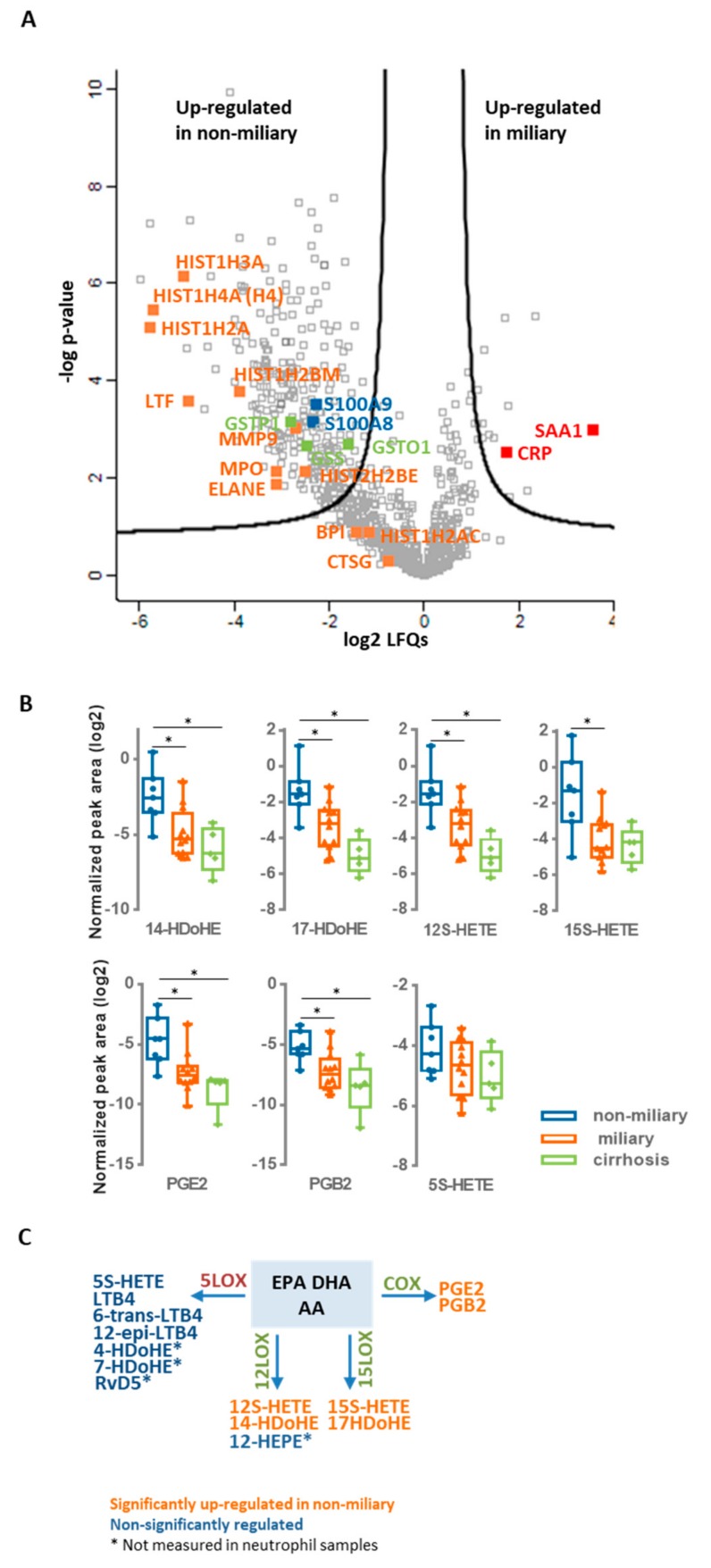
Untargeted multi-omics analysis of ascites samples. (**A**) Proteomics. The results of comparative proteomics analysis between miliary and non-miliary ascites samples are represented as volcano plot. Proteins above black lines were significantly regulated with fold changes > 2 and a fold discovery rate (FDR) < 0.05. NETs proteins are labeled in orange, whereas blue and red labeled proteins are related to local and systemic inflammation, respectively. Selected proteins linked to oxidative stress are labeled in green. (**B**) Eicosadomics. Boxplots show abundance levels of significantly regulated eicosanoids in non-miliary in comparison to miliary ascites samples (logarithmic scale with base 2; adjusted *p*-value < 0.05). * indicate *p*-value < 0.05. (**C**)Eicosanoid class switching. Three main precursors of eicosanoids are shown, arachidonic acid (AA), eicosapentaenoic acid (EPA), and docosahexaenoic acid (DHA), as well as eicosanoids, grouped regarding their biosynthetic processing through 5-LOX, 12-LOX, 15-LOX, or COX.

**Figure 2 cancers-12-00505-f002:**
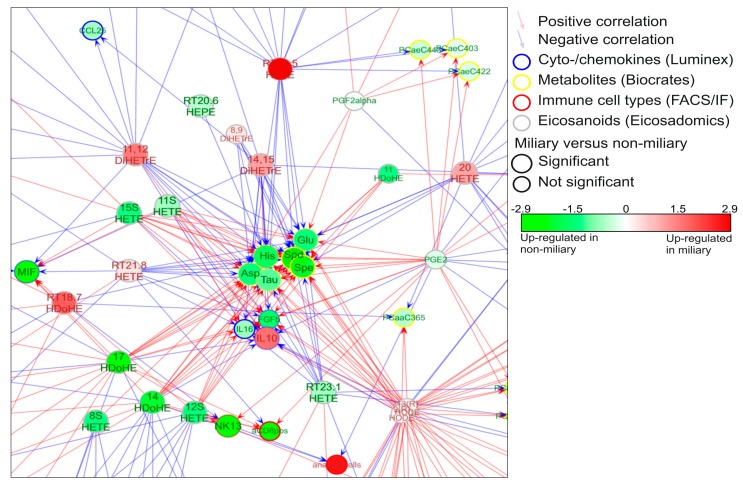
Network signature. A co-association network was built with molecules (eicosanoids, metabolites, proteins, RNA) and immune cells which were found significantly regulated in non-miliary compared to miliary ascites samples. Six metabolites (aspartate (Asp), glutamate (Glu), taurine (Tau), histamine (His), spermidine (Spd), spermine (Spe)) built a hub in the middle of the network and were found strongly regulated by eicosanoids. The complete network signature is represented in Appendix A.

**Figure 3 cancers-12-00505-f003:**
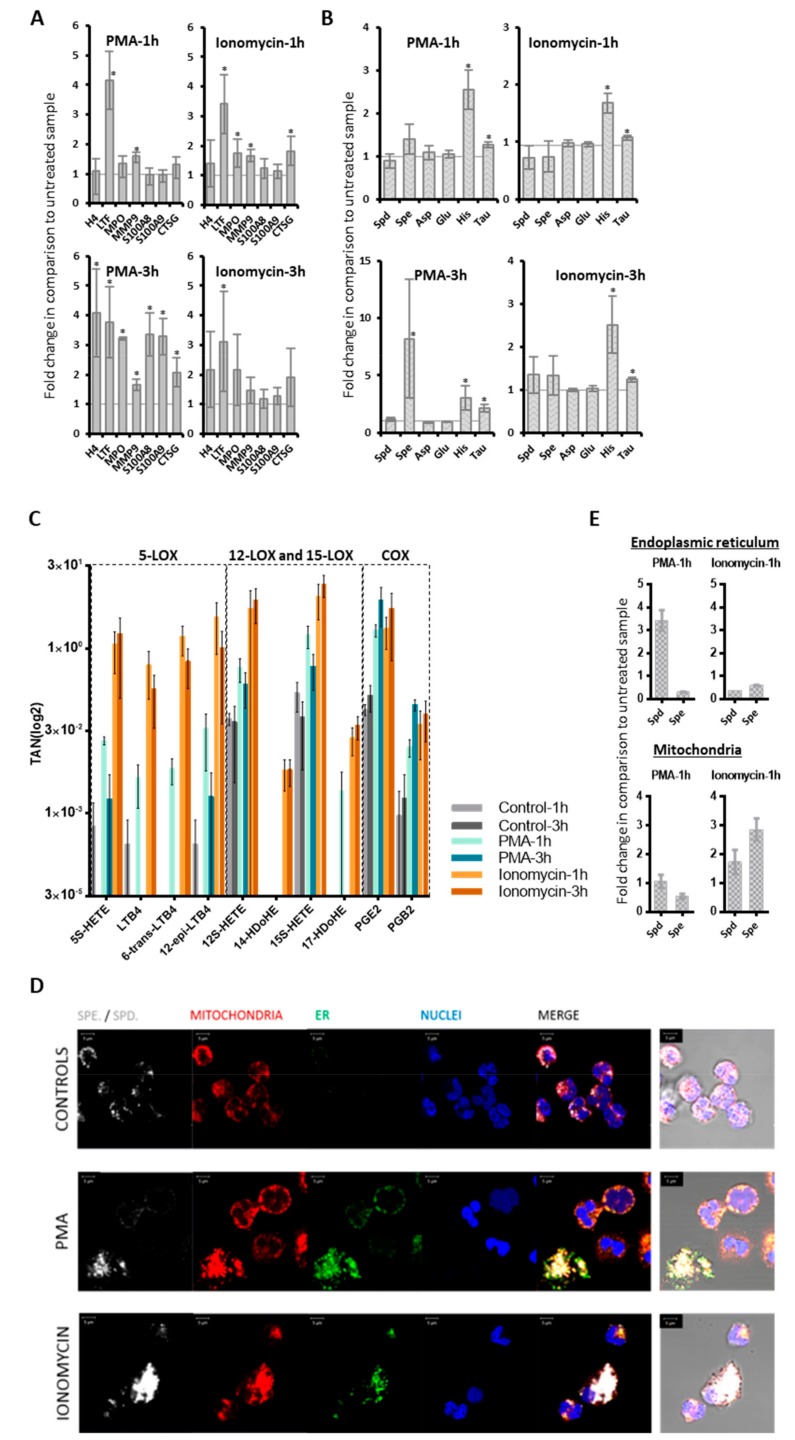
Targeted multi-omics analysis of neutrophils. Neutrophils isolated from healthy donors (n = 3) were treated with PMA (25 nM) or ionomycin (4 µM). The cell supernatants were isolated and analyzed. Error bars indicate standard deviations. * indicate *p*-value < 0.05. (**A**) Proteomics. The fold changes of NETs proteins and S1008/9 proteins in treated versus untreated samples are shown. (**B**) Metabolomics. The fold changes of six metabolites in treated versus untreated samples are represented. (**C**) Eicosadomics. The abundances of selected eicosanoids as total areas normalized to global deuterated standards (TAN) are blotted. Neutrophils (n = 3) were cultured in RPMI medium supplemented with 0.01% FCS. At all treatment conditions, eicosanoids were significantly up-regulated when comparing treated samples with their respective untreated control samples. (**D**) Immunolocalization of spermine/spermidine in neutrophils. Comparison between neutrophils which were untreated CONTROLS or activated for one hour with PMA or IONOMYCIN. Spermine/spermidine (Spe/Spd) is depicted in white, mitochondria, through staining of mitochondrial import receptor subunit TOM20 homolog (TOMM20), in red, and endoplasmic reticulum (ER) in green. Nuclei are counterstained in blue with DAPI. Merged images are provided as fluorescence confocal images with or without transmitted light channel for visualization of cellular morphology in addition to the structures of interest. Scale bars stand for 5 µm. (**E**) Levels of spermidine (Spd) and spermine (Spe) in mitochondria and endoplasmic reticulum (ER). Mitochondria and ER were isolated by sucrose gradient from lysed treated and untreated neutrophils, respectively. Levels of Spd and Spe were determined in these organelles and compared between treated and untreated cells, as represented.

**Figure 4 cancers-12-00505-f004:**
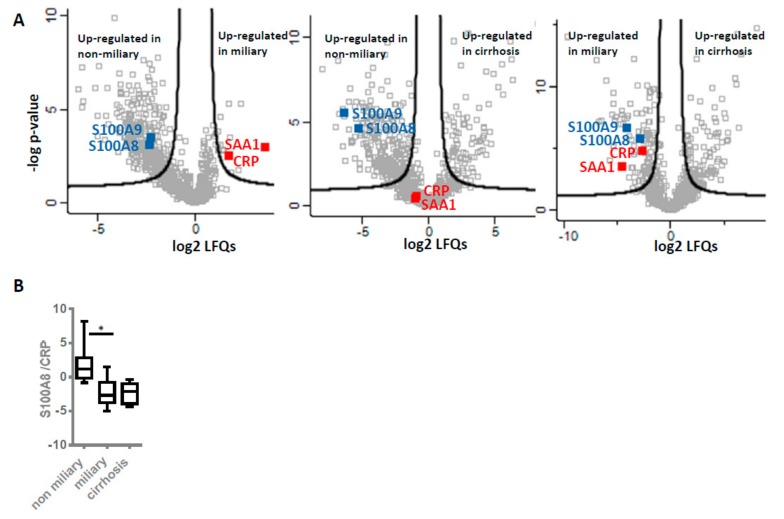
Inflammation marker expression levels and ratios. (**A**) Relative expression levels. Volcano plots illustrate results of comparative analysis using shotgun proteomics data of three groups of ascites patients: Non-miliary vs. miliary (left), non-miliary vs. cirrhosis (middle) and miliary vs. cirrhosis (right). Selected proteins usually related to local (blue) and systemic (red) inflammation are marked. Proteins above black lines are significantly regulated with fold changes > 2 and an FDR < 0.05. (**B**) Ratios. The abundance ratios of S100A8 to CRP within each patient group, as obtained from shotgun analyses, are shown.

**Figure 5 cancers-12-00505-f005:**
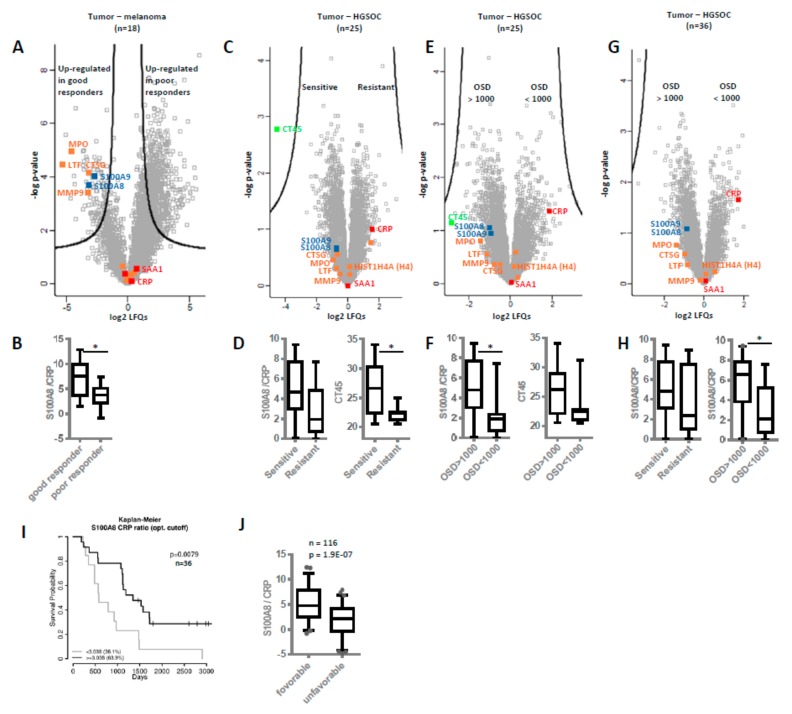
Neutrophil extracellular traps (NETs) proteins and S100A8/CRP ratio as prognostic biomarkers. Based on published proteomics data, levels of NETs proteins and S100A8/CRP ratios were compared in different patient cohorts. (**A**,**B**) Melanoma study by Zila et al. [49]. Differences in protein abundances as well as in S100/CRP ratios in cerebral metastases samples of good versus poor melanoma patient responders to MAPKi treatment are shown. For all volcano plots, proteins above black lines represent significantly regulated proteins, according to the original publications. Several neutrophil-specific proteins are labeled in orange, proteins related to local and systemic inflammation in blue and red, respectively. (**C**,**D**) HGSOC study by Coscia et al. [50]. Differences in protein abundances as well as in S100/CRP ratios between chemotherapy-sensitive and resistant HGSOC patients are shown. Additionally, the relative abundance levels of CT45 protein (cancer/testis antigen 45) are shown. (**E**,**F**) Same study as in C and D, re-grouping the HGSOC patients according to overall survival in OSD < 1000 and OSD > 1000. Again, the relative abundance levels of CT45 protein are shown. (**G**,**H**) Combining data from the study by Coscia *et al.* [50] and a study by Eckert et al. [51] In H, S100/CRP ratios grouping patient in sensitive/resistant or OSD>1000/<1000 are shown. (**I**) Kaplan–Meier analysis of survival probability based on S100A8/CRP abundance ratio. In tissue samples of 36 HGSOC patients, an optimal cutoff of 3.038 for the S100A8/CRP abundance ratio was determined. Patients with an S100A8/CRP abundance ratio higher or lower than the cutoff value are compared. **J**—Boxplots showing the distribution of the S100A8/CRP abundance ratio combining data from six independent studies. See Appendix A for detailed information about patients and samples included in this dataset. PFS—progression-free survival. OSD—overall survival days. * *p*-value < 0.05.

**Figure 6 cancers-12-00505-f006:**
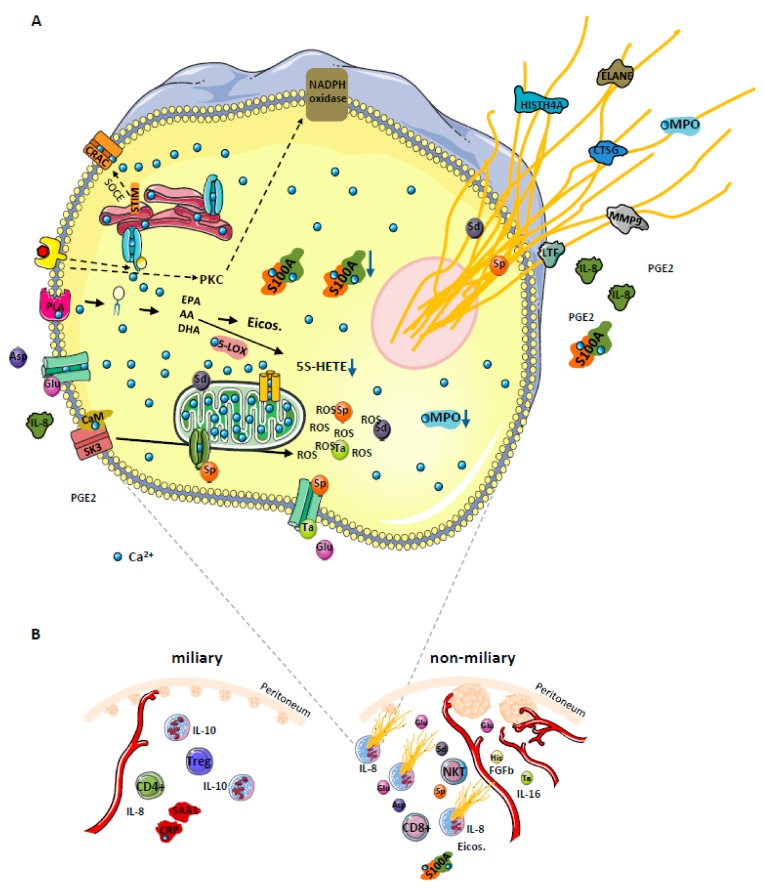
(**A**) Proposed model for NOX-independent NETs formation in high grade serous ovarian cancer (HGSOC) patients. Increase of cytosolic Ca^2+^ concentration leads to activation and translocation of several calcium-dependent and calcium-binding proteins, thus inducing hydrolysis of plasma membrane lipids (releasing PUFAs AA, EPA, and DHA) via activation of a calcium-dependent PLA. Among enzymes which metabolize these PUFAs into eicosanoids (5-LOX, 12-LOX, 15-LOX, COX and CYP450) only 5-LOX binds Ca^2+^. Under conditions of prolongated high intracellular Ca^2+^ concentration, the activity of 5-LOX enzyme is decreased, resulting in eicosanoid class switching process, exemplified by 5S-HETE decrease. Additionally, elevated Ca^2+^ levels promote translocation of calmodulin to SK3 receptors imbedded in plasma membranes, inducing receptor activation and induction of ROS production from mitochondria, resulting in NETosis. The S100A8/9 protein complex is released with the NETs. The sustained Ca^2+^ influx in the cell affects mitochondrial function and may initiate apoptosis. To attenuate this effect, the permeability transition pore channel and Ca^2+^ entry channels get closed by spermine (Sp). At the same time, the cell may use all three metabolites (spermine Sp, spermidine Sd, and taurine Ta) as ROS scavenger to deal with increased oxidative stress. Both positively charged polyamines stabilize DNA strands, and thus get released together with the NETs. We postulate that glutamine (Glu), released from cancer cells under hypoxic conditions, may induce Ca^2+^ influx in neutrophils by the activation of specific membrane receptors. Glutamine may also promote neutrophil activation by inducing the secretion of IL-8 and PGE2. Whereas IL-8 is a well-known chemoattractant and inducer of NETosis, PGE2 can induce eicosanoid class switching as observed in ascites samples. The dotted arrows indicate NOX-dependent NETosis and additional pathways via store-operated calcium entry (SOCE) promoting intracellular calcium mobilization. (**B**) Strong correlation between NETs formation, angiogenesis and the type of tumor spread in HGSOC. Selected proteins, eicosanoids, metabolites, immune cells, and processes are depicted as apparently regulated in miliary or non-miliary ascites samples. Strongly activated neutrophils, most probably by shedding millet-like and freshly build small tumor nodules, promote building of bigger but fewer tumor nods in the non-miliary type. In miliary spreading tumors, up-regulated IL-10 inhibits NETs formation. Additionally, neutrophils, depending on their activation status, modulate the immune system by determining the immune cell composition in the tumor microenvironment. Otherwise, increased angiogenesis associated with increased blood supply may contribute to less suppressive effects on neutrophils activation in the non-miliary type. PLA—phospholipase A; Asp—aspartate, His—histamine; calcium release-activated channel (CRAC); PKC—protein kinase C; FGFb—basic fibroblasts growth factor; Eicos.—eicosanoids; NTK—natural killer T cells; Treg—regulatory T cells.

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
