# Peer review of "Neutrophil Extracellular Trap Formation Correlates with Favorable Overall Survival in High Grade Ovarian Cancer"

_cancers, 2020, doi:10.3390/cancers12020505_

Round 1

Reviewer 1 Report

The manuscript by Muqaku is interesting. However, it is very difficult to understand due to the low quality of the figures.

Several points need to address properly:

The quality of figures is low. It is very difficult to look at the figures and understand the key finding. Importantly, the quality of figures from supplement data is better than in the in the main data. Please refer to this and change the data accordingly. Statistics is missing. At some figures, there are statistics calculated (fig 4, fig 5, fig S4b) but most of the others not. Please change accordingly. If there is non-significant, please also indicate. Fig 1B, color code would be nice to see and understand. Green box was appeared in 1B-12-HETE. Fig 2: Network signature is not able to read. Please change. Materials and methods part: Methods completely missing and was in the suppl. Please put few key methods to this part.

Author Response

Reviewer 1:
The manuscript by Muqaku is interesting. However, it is very difficult to understand due to the low quality of the figures.

We apologize for the low resolution which was chosen to keep the file size small. You will find a massively improved resolution in the revised version

Several points need to address properly:
The quality of figures is low. It is very difficult to look at the figures and understand the key finding. Importantly, the quality of figures from supplement data is better than in the in the main data. Please refer to this and change the data accordingly.

We apologize for the low resolution which was chosen to keep the file size small. You will find a massively improved resolution in the revised version

Statistics is missing. At some figures, there are statistics calculated (fig 4, fig 5, fig S4b) but most of the others not. Please change accordingly.

We have now included additional statements regarding statistical evaluations and included also indications thereof in Figures 1B and 3A. Please understand we omitted such indications in Figure 3C as there would have been too many asterisks making the figure illegible eventually. Instead, we have clearly indicated the significant events in the Legends to the Figure.

If there is non-significant, please also indicate.

Not significant events are now clearly visible as such.

Fig 1B, color code would be nice to see and understand. Green box was appeared in 1B-12-HETE.

We thank the reviewer for this helpful comment and now used such color code in the revised version

Fig 2: Network signature is not able to read. Please change.

We thank the reviewer for this helpful comment and now created a zoom in to make sure all labels are readable.

Materials and methods part: Methods completely missing and was in the suppl. Please put few key methods to this part.

We thank the reviewer for this helpful comment and complemented this section as requested.

Reviewer 2 Report

This study by Muqaku B et al, highlights the potential contribution of neutrophils to mechanisms of tumor spread and overall survival in HGSOC patients. The paper is straightforward, well written, and concise. Definitely deserves to be published and is a valuable contribution to the “cancers” journal. Some minor flaws need to be addressed before publication.

Minor points:

[1] Discussion, Lines 290-293:

“An active and relevant role of neutrophils for tumorigenesis has been recognized with regard to several tumors [53]. However, whether neutrophils rather promote cancer development or contribute to immune reactions inhibiting cancer growth, or whether neutrophils subtypes [54] may account for conflicting data, is still a matter of debate.” 

At that point, please make a comment about the predictive value of various blood cell ratios in colorectal cancer has been supported by several studies. Indeed, neutrophil–lymphocyte ratio indicates the balance between the inflammatory response and the antitumor immunity.

Relevant reference: Boussios S, et al. The Developing Story of Predictive Biomarkers in Colorectal Cancer. J Pers Med. 2019 Feb 7;9(1). pii: E12.

[2] Discussion, Lines 336-339:

“A substantial challenge for ovarian cancer cells in the peritoneum is caused by hypoxia [57] calling for metabolic adaptation [58]. Actually increased incidence of cell death may be associated with stress resulting in the release of DAMPs and the establishment of an adaptive immune response [59].”

Please, make here a comment from the therapeutic perspective. Importantly, hypoxia induces the down-regulation of BRCA1 expression, involved in DNA repair, cell cycle checkpoint control, and transcriptional regulation. This down-regulation is associated with a functional decrease in homologous recombination activity in hypoxic cells. In clinical practice, we aim to reduce overlapping toxicities by combined PARP inhibitors separately with several agents, including anti-angiogenics, immune checkpoint inhibitors, phosphoinositide 3-kinase (PI3K), protein kinase B (AKT), mammalian target of rapamycin (mTOR), WEE1, mitogen-activated protein kinase (MEK), and cyclin dependent kinase (CDK) 4/6 inhibitors.

Relevant reference: Boussios S, et al. Combined Strategies with Poly (ADP-Ribose) Polymerase (PARP) Inhibitors for the Treatment of Ovarian Cancer: A Literature Review. Diagnostics (Basel). 2019 Aug 1;9(3). pii: E87.

[3] Discussion, Lines 365-366:

“This may be due to a modulation of the adaptive immune system via recruitment of CD8+ T cells and suppression of regulatory T cells, which were found elevated in miliary samples (Fig. 2).”

In high-grade serous ovarian cancer, tumors harboring BRCA1/2 mutations demonstrated a higher neoantigen burden, and CD3+ / CD8+ tumor-infiltrating lymphocytes. Therapeutically, increased levels of PD-1 and PD-L1 expression on tumor-infiltrating immune cells as compared to homologous recombination proficient tumors indicates that PD-1/PD-L1 inhibitors have a better efficacy in BRCA1/2-mutated high-grade serous ovarian cancer.

Relevant reference: Boussios S, et al. PARP Inhibitors in Ovarian Cancer: The Route to "Ithaca". Diagnostics (Basel). 2019 May 18;9(2). pii: E55.

Author Response

Reviewer 2:
This study by Muqaku B et al, highlights the potential contribution of neutrophils to mechanisms of tumor spread and overall survival in HGSOC patients. The paper is straightforward, well written, and concise. Definitely deserves to be published and is a valuable contribution to the “cancers” journal. Some minor flaws need to be addressed before publication.

We highly appreciate the helpful comments of this reviewer which helped to substantially improve our manuscript. All points raised below have been fully addressed in the revised version including all indicated references.

Minor points:

[1] Discussion, Lines 290-293:

“An active and relevant role of neutrophils for tumorigenesis has been recognized with regard to several tumors [53]. However, whether neutrophils rather promote cancer development or contribute to immune reactions inhibiting cancer growth, or whether neutrophils subtypes [54] may account for conflicting data, is still a matter of debate.”

At that point, please make a comment about the predictive value of various blood cell ratios in colorectal cancer has been supported by several studies. Indeed, neutrophil–lymphocyte ratio indicates the balance between the inflammatory response and the antitumor immunity.

Relevant reference: Boussios S, et al. The Developing Story of Predictive Biomarkers in Colorectal Cancer. J Pers Med. 2019 Feb 7;9(1). pii: E12.

[2] Discussion, Lines 336-339:

“A substantial challenge for ovarian cancer cells in the peritoneum is caused by hypoxia [57] calling for metabolic adaptation [58]. Actually increased incidence of cell death may be associated with stress resulting in the release of DAMPs and the establishment of an adaptive immune response [59].”

Please, make here a comment from the therapeutic perspective. Importantly, hypoxia induces the down-regulation of BRCA1 expression, involved in DNA repair, cell cycle checkpoint control, and transcriptional regulation. This down-regulation is associated with a functional decrease in homologous recombination activity in hypoxic cells. In clinical practice, we aim to reduce overlapping toxicities by combined PARP inhibitors separately with several agents, including anti-angiogenics, immune checkpoint inhibitors, phosphoinositide 3-kinase (PI3K), protein kinase B (AKT), mammalian target of rapamycin (mTOR), WEE1, mitogen-activated protein kinase (MEK), and cyclin dependent kinase (CDK) 4/6 inhibitors.

Relevant reference: Boussios S, et al. Combined Strategies with Poly (ADP-Ribose) Polymerase (PARP) Inhibitors for the Treatment of Ovarian Cancer: A Literature Review. Diagnostics (Basel). 2019 Aug 1;9(3). pii: E87.

[3] Discussion, Lines 365-366:

“This may be due to a modulation of the adaptive immune system via recruitment of CD8+ T cells and suppression of regulatory T cells, which were found elevated in miliary samples (Fig. 2).”

In high-grade serous ovarian cancer, tumors harboring BRCA1/2 mutations demonstrated a higher neoantigen burden, and CD3+ / CD8+ tumor-infiltrating lymphocytes. Therapeutically, increased levels of PD-1 and PD-L1 expression on tumor-infiltrating immune cells as compared to homologous recombination proficient tumors indicates that PD-1/PD-L1 inhibitors have a better efficacy in BRCA1/2-mutated high-grade serous ovarian cancer.

Relevant reference: Boussios S, et al. PARP Inhibitors in Ovarian Cancer: The Route to "Ithaca". Diagnostics (Basel). 2019 May 18;9(2). pii: E55.